# Learning from Imperfect Annotations: An End-to-End Approach

## Abstract

Many machine learning systems today are trained on large amounts of human-annotated data. Annotation tasks that require a high level of competency make data acquisition expensive, while the resulting labels are often subjective, inconsistent, and may contain a variety of human biases. To improve data quality, practitioners often need to collect multiple annotations per example and aggregate them before training models. Such a multi-stage approach results in redundant annotations and may often produce imperfect "ground truth" labels that limit the potential of training supervised machine learning models. We propose a new end-to-end framework that enables us to: (i) merge the aggregation step with model training, thus allowing deep learning systems to learn to predict ground truth estimates directly from the available data, and (ii) model difficulties of examples and learn representations of the annotators that allow us to estimate and take into account their competencies. Our approach is general and has many applications, including training more accurate models on crowdsourced data, ensemble learning, as well as classifier accuracy estimation from unlabeled data. We conduct an extensive experimental evaluation of our method on 5 crowdsourcing datasets of varied difficulty and show accuracy gains of up to 25% over the current state-of-the-art approaches for aggregating annotations, as well as significant reductions in the required annotation redundancy. We further conduct an ablation study to evaluate the effect of both end-to-end learning and instance features and show that both contribute to the performance gains achieved by the proposed method.

## 1 Introduction

The rising popularity and recent success of deep learning has resulted in machine learning systems that rely on large amounts of annotated training data (LeCun et al., 2015; Wu et al., 2016; Gulshan et al., 2016; Esteva et al., 2017). The most common, scalable way to collect such large amounts of training data is through crowdsourcing (Howe, 2006). Crowdsourcing works well in simple settings where annotation tasks do not require domain expertise—for example, in object detection and recognition tasks in natural images and videos (e.g., Deng et al., 2009; Kovashka et al., 2016). However, annotation in specialized domains such as medical pathology requires a certain level of competency and expertise from the annotators which makes annotation expensive. Moreover, often times there is high rate of disagreement even between experts, which results in increasingly subjective and inconsistent labels (Elmore et al., 2015; Hutson et al., 2019).

A typical approach to dealing with subjectivity is to treat each annotation as simply noisy, collect multiple redundant labels per example (e.g., from different annotators), and then aggregate them using majority voting or other more advanced techniques (e.g., Dawid & Skene, 1979; Carpenter, 2008; Liu et al., 2012; Bachrach et al., 2012; Zhou et al., 2015; Zhou & He, 2016) to obtain a single "ground truth" label. At the expense of redundancy, this results in better data quality and more accurate estimates of the ground truth. More recently, the emerging systems for *data programming* and *weak supervision* also internally rely on label aggregation techniques similar to methods used for solving the crowdsourcing problem. Snorkel (Ratner et al., 2017; Bach et al., 2019) is a popular such system and was designed for efficient and low-cost creation of large-scale labeled datasets using programmatically generated, so-called *weak labels*. However, as we show in our empirical evaluation none of these systems solve label aggregation effectively in the presence of high subjectivity. We argue that to become more effective, these methods need to make use of meta-data and other types of information that may be available about the data instances and the annotators labeling them.

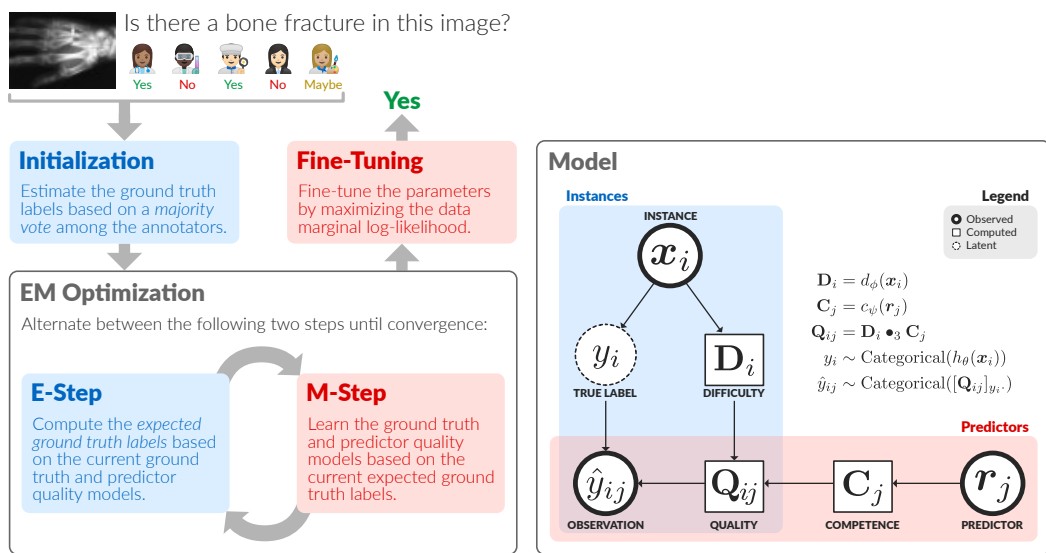

**Figure 1:** Overview of the proposed algorithm and probabilistic model.

To this end, we propose a novel approach that allows us to train accurate predictive models of the ground truth *directly on the non-aggregated imperfectly labeled data*. Our method merges the two steps of: (i) aggregating subjective, weak, or noisy annotations, and (ii) training machine learning models. At training time, along with learning a model that predicts the ground truth, we also learn models of the difficulty of each example and the competence of each annotator in a generalizable manner (i.e., these models can make predictions for previously unseen examples and annotators). Our approach can be effectively used for training on crowdsourced data as well as on weakly labeled data, and also be used within frameworks such as Snorkel (Ratner et al., 2017; Bach et al., 2019) and significantly improve their performance. We propose a method that can:

1. **Learn truth estimators:** Learn functions representing the underlying ground truth, while imposing almost no constraints (as opposed to prior work). In fact, we are able to leverage the capacity of deep neural networks along with the interpretability provided by Bayesian models, in order to obtain highly expressive estimators of the underlying truth.

2. **Learn quality estimators:** Learn functions that estimate the quality of each annotator. When annotators can be described by some features (e.g., age, gender, location, etc. of an Amazon Mechanical Turk annotator, instead of just an ID), our quality estimators are able to generalize to new, previously unseen, predictors. Previous work only considered estimating accuracies of a fixed set of predictors, without being able to leverage any information we might have about them. Furthermore, in contrast to previous work, we are also able to predict the per-instance predictor comptencies (i.e., our method can determine whether a human annotator is an expert for a subset of queries, instead of just estimating his/her overall accuracy), which is done by learning dependencies between the instances and the annotators. Finally, our approach is able to distinguish between multiple different types of errors by estimating the full confusion matrix for each instance-predictor pair.

3. **Be easily extended:** The truth and quality estimators can take arbitrary functional forms and fully leverage the expressivity of deep neural networks.

Both human annotators and machine learning classifiers may sometimes be unable to make predictions about certain aspects of the ground truth (e.g., human annotators may be unsure about what the correct answer to a question). The proposed method is formulated in a way that allows it to be extended such that it can also *learn decision function estimators* for the annotators (i.e., estimators that predict whether an annotator will be able to provide a prediction for a given data instance). These estimators can have significant implications for data annotation systems where the cost of querying annotators is high (e.g., when these annotators are highly qualified, such as doctors or other kinds of domain experts). This is because it allows for better matching annotators to instances, thus reducing the required amount of annotation redundancy. An overview of the proposed approach and model is shown in Figure 1, and a detailed description is provided in Section 3.

## 2 RELATED WORK

Research on label aggregation and crowdsourcing dates back to the early 1970s, when Dawid & Skene (1979) proposed a probabilistic model to estimate ground truth labels using the expectation maximization (EM) algorithm. Since then, a variety of generalizations and improvements upon the original method have been proposed (Whitehill et al., 2009; Welinder et al., 2010; Liu et al., 2012; Zhou et al., 2015; Zhou & He, 2016). One of the central parts of the label aggregation algorithms is estimation of the accuracy of the annotators (or predictors) without having access to the ground truth. This problem has been of independent interest to the machine learning community, was termed as *estimating accuracy from unlabeled data* and studied by Collins & Singer (1999), Dasgupta et al. (2001), Bengio & Chapados (2003), Madani et al. (2004), Schuurmans et al. (2006), Balcan et al. (2013), and Parisi et al. (2014), among others. Almost none of the previous approaches explicitly considered modeling the ground truth, but rather assumed either some form of independence or knowledge of the true label distribution.

Collins & Huynh (2014) reviewed many methods that were proposed for estimating the accuracy of medical tests in the absence of a gold standard. Platanios et al. (2014) proposed formulating the problem as an optimization problem that uses agreement rates between multiple noisy annotators. Platanios et al. (2017) improved upon agreement-based accuracy estimation using logical constraints between the noisy labels. Tian & Zhu (2015) proposed a max-margin majority voting scheme applied to crowdsourcing. More recently, Khetan et al. (2017) proposed to use a parametric function to model the ground truth and showed that the approach can sometimes be functional even in the limit of a single noisy label per example. Among recent approaches, Zhou et al. (2015) formulated the problem as a form of regularized minimax conditional entropy and established one of the most competitive baselines on many public crowdsourcing datasets.

Our proposed method generalizes the approaches of Zhou et al. (2015), Platanios et al. (2016), and Khetan et al. (2017). Similar to Platanios et al. (2016), we define a generative process for our observations. However, our model is able to handle categorical labels, as opposed to just binary. Similar to Zhou et al. (2015), we define the confusion matrix for each instance-predictor pair as a function of instance difficulty and predictor competence.[1] However, we explicitly learn the difficulty and competence as functions, which allows us to generalize to previously unseen instances and annotators. Interestingly, the inference algorithm for our generative probabilistic model has a similar form to that of Zhou et al. (2015) (except for the explicit learning of the ground truth, difficulty, and competence functions). Thus, we also show that the algorithm of Zhou et al. (2015) can be re-derived as an EM inference algorithm for a generative model, simplifying the argument of the original paper. Finally, similar to Khetan et al. (2017), we use a parametric function to model the ground truth, and also go a step further and propose to use parametric functions to model the instance difficulties and predictor competences. Thus, our approach enables estimation of which annotators are likely to perform better on which instances, potentially enabling more optimal allocation of annotators and thus annotation cost reductions.

In contrast to prior work, our method also allows for *end-to-end learning*. Prior methods do not allow for this as they (implicitly) separate ground truth inference (i.e., label aggregation) from model training. More specifically, previously one would have to train a machine learning model in two stages: (i) infer the ground truth labels from the provided annotations, and (ii) train a machine learning model on the inferred labels. Our approach merges these two stages and allows us to train machine learning models directly on the imperfect annotations. In Section 4.3, we conduct an ablation study that showcases the performance gains obtained by employing end-to-end learning in this fashion.

## 3 PROPOSED METHOD

We denote the observed data by $\mathcal{D} = \{\boldsymbol{x}_i, \hat{\mathcal{Y}}_i\}_{i=1}^N$, where $\hat{\mathcal{Y}}_i = \{\mathcal{M}_i, \{\hat{y}_{ij}\}_{j \in \mathcal{M}_i}\}$, $\mathcal{M}_i$ is the set of predictors that made predictions for instance $\boldsymbol{x}_i$, and $\hat{y}_{ij}$ is the output of predictor $\hat{f}_j$ for instance $\boldsymbol{x}_i$. Our goal is to learn functions representing the underlying ground truth and predictor qualities, given our observations $\mathcal{D}$.

---

[1]The idea of modeling instance difficulties and annotator competencies has been studied before by Carpenter (2008) and Bachrach et al. (2012), among others.

**Ground Truth.** We define the ground truth as a function $h_\theta(\boldsymbol{x}_i)$ that is parameterized by $\theta$ and that approximates the true distribution of the label given $\boldsymbol{x}_i$. In our setting, $h_\theta(\boldsymbol{x}_i) \in \mathbb{R}^C_{\geq 0}$ and $\sum_j [h_\theta(\boldsymbol{x}_i)]_j = 1$, where $C$ is number of values the label can take (i.e., assuming categorical labels). More specifically, $[h_\theta(\boldsymbol{x}_i)]_k \triangleq \mathbb{P}(y_i = k \mid \boldsymbol{x}_i)$, where we use square brackets and subscripts to denote indexing of vectors, matrices, and tensors. For example, $h_\theta$ could be a deep neural network that would normally be trained in isolation using the cross-entropy loss function. In our method the network is trained using the Expectation-Maximization algorithm, as described in the next section.

**Predictor Qualities.** We define the predictor qualities as the confusion matrices $\mathbf{Q}_{ij} \in \mathbb{R}^{C \times C}_{\geq 0}$, for each instance $\boldsymbol{x}_i$ and predictor $\hat{f}_j$, where $\sum_l [\mathbf{Q}_{ij}]_{kl} = 1$, for all $k \in \{1, \ldots, C\}$. $[\mathbf{Q}_{ij}]_{kl}$ represents the probability that predictor $\hat{f}_j$ outputs label $l$ given that the true label of instance $\boldsymbol{x}_i$ is $k$. We define these confusion matrix in a way that generalizes the successful approach of Zhou et al. (2015):[2]

$$\mathbf{Q}_{ij} = \mathbf{D}_i \bullet_3 \mathbf{C}_j, \tag{1}$$

where $\bullet_i$ represents an inner product along the $i^{\text{th}}$ dimension of the two tensors, and:

- $\mathbf{D}_i = d_\phi(\boldsymbol{x}_i)$ represents the *difficulty* tensor for instance $\boldsymbol{x}_i$, where $d$ is a function parameterized by $\phi$, $\mathbf{D}_i \in \mathbb{R}^{C \times C \times L}$, and $L$ is a latent dimension (it is a hyperparameter of our model). $[\mathbf{D}_i]_{kl-}$ is an $L$-dimensional embedding representing the likelihood of confusing $\boldsymbol{x}_i$ as having label $l$ instead of $k$, when $k$ is its true label.
- $\mathbf{C}_j = c_\psi(\boldsymbol{r}_j)$ represents the *competence* tensor for predictor $\hat{f}_j$, where $c$ is a function parameterized by $\psi$, $\boldsymbol{r}_j$ is some representation of $\hat{f}_j$ (e.g., could be a one-hot encoding of the predictor, in the simplest case), and $\mathbf{C}_j \in \mathbb{R}^{C \times C \times L}$. $[\mathbf{C}_j]_{kl-}$ is an $L$-dimensional embedding representing the likelihood that predictor $\hat{f}_j$ confuses label $k$ for $l$, when $k$ is the true label.

Using $L > 1$ allows the instance difficulties and predictor competences to encode more information. An intuitive way to think about this is that we are embedding difficulties and competencies in a common latent space, which can be thought of as jointly clustering them. This is in fact very similar to how matrix factorization methods are used for collaborative filtering in recommender systems.

Our goal is to learn functions $h_\theta$, $d_\phi$, and $c_\psi$, given observations $\mathcal{D}$. In order to do that, we propose the following generative process for our observations. For $i = 1, \ldots, N$, we first sample the true label for $\boldsymbol{x}_i$, $y_i \sim \text{Categorical}(h_\theta(\boldsymbol{x}_i))$. Then, for $j \in \mathcal{M}_i$, we sample the predictor output $\hat{y}_{ij} \sim \text{Categorical}([\mathbf{Q}_{ij}]_{y_i -})$, where $[\mathbf{Q}_{ij}]_{y_i -}$ represents the $y_i^{\text{th}}$ row of $\mathbf{Q}_{ij}$. In the next section, we propose an algorithm learning the parameters $\theta$, $\phi$, and $\psi$.

## 3.1 Learning

A widespread approach for performing learning with probabilistic generative models, is to maximize the likelihood of the observed data with respect to the model parameters. Let $\boldsymbol{y} = \{y_i\}_{i=1}^N$. The complete likelihood of a single observation, $\hat{y}_{ij}$, can be derived as follows:

$$p(\mathcal{D}, \boldsymbol{y}) = \prod_{i=1}^N p(y_i) \prod_{j \in \mathcal{M}_i} p(\hat{y}_{ij} \mid y_i), \tag{2}$$

where $p(\hat{y}_{ij} \mid y_i)$ depends on $\mathbf{Q}_{ij}$. There are two main approaches in which we can use the likelihood function of Equation 2 for learning: (i) marginalize out the $y_i$ latent variables and then maximize with respect to $\theta$, $\phi$, and $\psi$, or (ii) use the expectation maximization (EM) algorithm originally proposed by Dempster et al. (1977). It has previously been observed that the EM algorithm can perform much better than approach (i) for mixture models (Bishop, 2006), as the latter tends to get stuck in bad local optima. Since our model resembles a Bernoulli mixture model with the latent assignments being defined by the $y_i$'s, we decided to use the EM algorithm.[3] The steps of the EM algorithm for our model are as follows:

---

[2]We also perform a normalization step such that all elements of $\mathbf{Q}_{ij}$ are non-negative and such that each row sums to 1 (thus making each row a valid probability distribution).

[3]In fact, we also experimented with the marginalization approach and it consistently underperformed EM.

**E-Step.** We need to compute the expectation of $y_i$ given $\mathcal{D}$ and $\boldsymbol{y}_{\setminus i}$ (which denotes all of $\boldsymbol{y}$ except for $y_i$), for all $i = 1, \ldots, N$, and we know that:

$$p(y_i \mid \mathcal{D}, \boldsymbol{y}_{\setminus i}) \propto p(\mathcal{D}, \boldsymbol{y}) = \prod_{s=1}^{N} p(y_s) \prod_{j \in \mathcal{M}_s} p(\hat{y}_{sj} \mid y_s). \tag{3}$$

Therefore, by removing all terms that do not depend on $y_i$ and normalizing, we obtain the following expectation (which we compute during this step, while keeping $\theta$, $\phi$, and $\psi$ fixed):[4]

$$\mathbb{E}_{\boldsymbol{y} \mid \mathcal{D}} \left\{ \mathbb{1}_{[y_i = k]} \right\} = p(y_i = k \mid \mathcal{D}, \boldsymbol{y}_{\setminus i}) = \frac{\lambda_i^k}{\sum_{l=1}^{C} \lambda_i^l}, \text{ where } \lambda_i^k = [h_\theta(\boldsymbol{x}_i)]_k \prod_{j \in \mathcal{M}_i} \frac{[\mathbf{Q}_{ij}]_{k\hat{y}_{ij}}}{\sum_{l=1}^{C} [\mathbf{Q}_{ij}]_{l\hat{y}_{ij}}}. \tag{4}$$

For brevity, in what follows, we use the following notation $\tilde{y}_i^k := \mathbb{E}_{\boldsymbol{y} \mid \mathcal{D}} \{ \mathbb{1}_{[y_i = k]} \}$.

**M-Step.** We maximize the following log-likelihood function with respect to $\theta$, $\phi$, and $\psi$, while using the values of $\tilde{y}_i^k$ computed in the last E-step:

$$\mathcal{L} = \prod_{i=1}^{N} p(y_i = \tilde{y}_i) \prod_{j \in \mathcal{M}_i} p(\hat{y}_{ij} \mid y_i = \tilde{y}_i) \Rightarrow \tag{5}$$

$$\log \mathcal{L} = \sum_{i=1}^{N} \log p(y_i = \tilde{y}_i) + \sum_{i=1}^{N} \sum_{j \in \mathcal{M}_i} \log p(\hat{y}_{ij} \mid y_i = \tilde{y}_i) \Rightarrow \tag{6}$$

$$\log \mathcal{L} = \sum_{i=1}^{N} \sum_{k=1}^{C} \tilde{y}_i^k \log[h_\theta(\boldsymbol{x}_i)]_k + \sum_{i=1}^{N} \sum_{k=1}^{C} \tilde{y}_i^k \sum_{j \in \mathcal{M}_i} \log \left[ \frac{[\mathbf{Q}_{ij}]_{k\hat{y}_{ij}}}{\sum_{l=1}^{C} [\mathbf{Q}_{ij}]_{l\hat{y}_{ij}}} \right]. \tag{7}$$

The training procedure for learning the parameters $\theta$, $\phi$, and $\psi$ consists of iterating over the E-step and the M-step shown above, until convergence, where convergence can be measured by computing the change in the parameter values across learning iterations. It is important to note that EM finds local optima of the likelihood function, and so the starting point can play an important role. Also, as Platanios et al. (2016) mention, there exists an inherent symmetry in our model that can be problematic. The likelihood of any observed data is the same if we flip the true underlying labels and the predictor qualities (i.e., set $y_i^{\text{flipped}} = 1 - y_i$ and $\mathbf{Q}_{ij}^{\text{flipped}} = 1 - \mathbf{Q}_{ij}$). We would like to somehow encode the prior assumption that most of the predictors are correct most of the time. One way to do this is by choosing the starting point of the EM algorithm carefully.

**Initialization.** In the E-step shown in Equation 4, we compute the expected values of the true underlying labels, $y_i$. We can encode the assumption mentioned in the previous paragraph by replacing the first E-step with a majority vote among the predictors:

$$\tilde{\mathbb{E}}_{\boldsymbol{y} \mid \mathcal{D}} \left\{ \mathbb{1}_{[y_i = k]} \right\} = \frac{\sum_{j \in \mathcal{M}_i} \mathbb{1}_{[\hat{y}_{ij} = k]}}{|\mathcal{M}_i|}, \tag{8}$$

where $|\mathcal{M}_i|$ denotes the size of set $\mathcal{M}_i$. We initialize the EM algorithm by replacing the first E-step with this majority vote approximation. As we show in our experiments, this helps us avoid the aforementioned symmetry, and thus we refer to this initialization scheme as *symmetry-breaking initialization*. Note also that in the case where the predictors provide us with $\mathbb{P}(\hat{y}_{ij} = k)$, instead of a single categorical value, we can still use this initialization scheme by replacing $\mathbb{1}_{[\hat{y}_{ij} = k]}$ with $\mathbb{P}(\hat{y}_{ij} = k)$, in Equation 8.

**Marginal Likelihood Fine-Tuning.** In our experiments we found that maximizing the marginal likelihood function after EM converges tends to improve performance. We refer to this step as *marginal likelihood fine-tuning*. More specifically, after the values of the parameters $\theta$, $\phi$, and $\psi$, converge to fixed values across multiple EM steps, we solve the following maximization problem using these fixed values as the initial point:

$$\max_{\theta, \phi, \psi} \sum_{\boldsymbol{y}} p(\mathcal{D}, \boldsymbol{y}) \quad \Leftrightarrow \quad \max_{\theta, \phi, \psi} \sum_{i=1}^{N} \sum_{j \in \mathcal{M}_i} \log \sum_{k=1}^{C} [h_\theta(\boldsymbol{x}_i)]_k \frac{[\mathbf{Q}_{ij}]_{k\hat{y}_{ij}}}{\sum_{l=1}^{C} [\mathbf{Q}_{ij}]_{l\hat{y}_{ij}}}. \tag{9}$$

---

[4]Note that $\mathbf{Q}_{ij}$ depends on $\phi$ and $\psi$.

## 3.2 Instance and Predictor Representations

A major advantage of the proposed approach over prior work is that we learn models of the ground truth and the predictor qualities as functions of some representations (i.e., representations of the data instances, $x_i$, and of the annotators, $r_j$). It is thus important to define these representations. For many problems, the representations of the data instances can be defined in the same manner as was previously done when performing supervised learning (e.g., we can directly use raw pixel values representing images). However, predictor representations are introduced here for the first time.[5] A simple approach would be to use a one-hot encoding of the predictors. However, this would not allow for any amount of information sharing across predictor (e.g., what if two predictors are very similar). We know from prior work that modeling dependencies between the predictors can be very important (e.g., Platanios et al., 2016). One way to allow for that is to learn vector embedding representations for the predictors, which would be implicitly equivalent to clustering them. Ideally, one would want to use any available information about these predictors (e.g., Amazon Mechanical Turk annotators could be described by their age, location, etc.). Unfortunately, we could not find any public datasets that provide such information about the predictors/annotators, and hence we used embedding representations in our empirical study.

## 3.3 Discussion

The approach we have proposed in this section can be thought of as introducing a new loss function for training the model $h_\theta$ using multiple imperfect labels per training instance, each coming from a different sources. This new loss function introduces latent variables representing the ground truth labels, as well as a couple of auxiliary models that are learned, and which represent the instance difficulties and predictor competences. We also proposed an EM-based algorithm to minimize this new loss function as well as an initialization scheme. Perhaps most interestingly, a key difference between this approach and previous work is that we are able to explicitly learn functions that output the likelihood that a predictor will label a specific instance correctly. This enables using this approach to perform crowdsourcing more actively by assigning annotators to instances they are more likely to label correctly, thus helping reduce redundancy and drive costs down.

## 3.4 Extending to Multi-Label Settings

Our method can easily be extended to handle settings where we have multiple categorical labels that can be assigned to each instance. In that case, the model per label is defined in the same way as previously, except that now the functions $h_\theta$, $d_\phi$, and $c_\psi$ also take as input a representation for the label (e.g., a label embedding). This allows us to share information across labels and can be thought of as a generalization of the approach by Platanios et al. (2016), where information is shared by clustering the labels. Furthermore, it allows us to use the proposed method in extreme classification settings (e.g., Prabhu & Varma, 2014) or settings where the number of labels is not fixed and known *a priori* and can keep increasing (e.g., face recognition; Weinberger & Saul, 2009; Liu et al., 2016). This is made possible by learning label representations and then letting the difficulty and competence functions also take as input a pairs of labels and return a vector instead of a three-dimensional tensor. In the next section, we show how learning label representations can significantly enhance the robustness and performance of our approach.

## 4 Experiments

We evaluate the proposed approach on multiple datasets from the crowdsourcing domain, all of which have ground truth labels, (multiple) subjective annotations for each example, as well as information on who provided each annotation (i.e., the annotator ID):

1. Blue Birds (BB) (Welinder et al., 2010): Bird photos labeled as *Indigo Bunting* or *Blue Grosbeak*.
2. Word Similarity (WS) (Snow et al., 2008): Pairs of words labeled as similar or dissimilar.
3. RTE (Snow et al., 2008): Pairs of sentences labeled as whether the first entails the second.
4. Medical Causes (MC) (Dumitrache et al., 2018): Sentences that contain 2 medical terms labeled if one of the terms *causes* the other (e.g., *pancreatic adenocarcinoma* causes *weight loss*).

---

[5]We note that learning representations of the agent behaviors has been recently explored in the context of imitation and reinforcement learning (e.g., Grover et al., 2018).

**Table 1:** Statistics for the datasets we used in our experiments. "**#Predictors**" refers to the total number of predictors in the dataset, "**Average Redundancy**" refers to the average number of predictions provided for each instance, "**Average Accuracy**" refers to the average predictor accuracy, and "**Random Accuracy**" refers to the accuracy obtained of a completely random predictor.

| Dataset | #Instances | #Predictors | Average Redundancy | Average Accuracy (%) | Random Accuracy (%) |
|---|---|---|---|---|---|
| BLUE BIRDS (Welinder et al., 2010) | 108 | 39 | 39 | 63.56 | 50.00 |
| WORD SIMILARITY (Snow et al., 2008) | 30 | 10 | 10 | 81.33 | 50.00 |
| RTE (Snow et al., 2008) | 800 | 164 | 10 | 84.13 | 50.00 |
| MEDICAL CAUSES (Dumitrache et al., 2018) | 3,984 | 408 | 15 | 32.40 | 7.00 |
| MEDICAL TREATS (Dumitrache et al., 2018) | 3,984 | 408 | 15 | 38.88 | 7.00 |

5. Medical Treats (MT) (Dumitrache et al., 2018): The same sentences with medical terms labeled if one of the terms *treats* the other (e.g., *aspirin* treats *pain*).

The last two datasets are in fact part of a single dataset on medical relations, and thus we are able to perform experiments both with the single task formulation of our algorithm and the multi-task formulation. As we discuss at the end of this section, this allows us to show how our approach can be used to share information across labels and improve the quality of the learned models.

Statistics for these datasets are provided in Table 1. Note that these datasets were provided without the associated features of the annotator identifiers and thus we are unable to evaluate the usefulness of annotator features (we instead learn embeddings for the annotators). Unfortunately we were unable to obtain any crowdsourcing datasets with associated annotator meta-data. This is probably due to the fact that no prior method is able to make use of such information. However, we do make use of instance features for all of these datasets. In cases where such features were not readily available, we manually computed them by using pre-trained machine learning models. We are making all such features and annotator identification information publicly available in a standardized format. More details are provided in our code and data repository which is available at `http://anonymous`.

## 4.1 EXPERIMENTAL SETUP

We perform experiments using the following two variants of our approach:

– **LIA**: A version of our method which uses instance and predictor features specific to each dataset. When features are not available for the instances and/or the predictors, we learn embeddings of size 16 which are initialized randomly and optimized along with the other model parameters during the M-step (see Section 3.1).
– **LIA-ML**: A multi-label variant of the aforementioned method. This method is only used with the medical relations datasets. In this case, we consider all 14 medical relations included in the dataset jointly and only evaluate on the two for which the ground truth is provided (i.e., "causes" and "treats"). We use this method variant in order to show how our approach can effectively share information across labels.

In both instances of **LIA**, $h_\theta$ and $d_\phi$ are multi-layer perceptrons (MLPs) with 4 layers of 16 hidden units each, with the only exception being the medical relations dataset where we used 32 units for each layer. $c_\psi$ is always modeled as a linear function. Note that for both the embedding sizes and the MLP sizes, we did not perform an extensive search to choose these values; we rather performed a small grid search and selected the number that resulted in the highest validation data likelihood. We compare against the following baselines for ground truth estimation:

– **MAJ**: Simple majority voting. Note that we use *soft* majority voting whenever possible, i.e., we use soft labels (probabilities or confidence scores) whenever the predictors provide them, instead of always thresholding them to obtain discrete labels.
– **MMCE**: Regularized minimax conditional entropy by Zhou et al. (2015), which has been shown to outperform alternatives. We consider it the current state-of-the-art for crowdsourcing.
– **Snorkel**: A method originally designed for aggregating annotations of programmatic weak predictors proposed by Ratner et al. (2017), which is part of a popular software package that allows for subsequent training of machine learning models on the aggregated data.
– **MeTaL**: Successor to Snorkel, proposed by Ratner et al. (2018). For both this method and for **Snorkel** we use the original implementation provided by the authors.[6]

---

[6] `https://github.com/HazyResearch/snorkel`.

**Table 2:** Accuracy across varying levels of redundancy, for all datasets we used in our experiments. For each experiment, we report mean accuracy and standard error over 50 runs from random initializations. The best results are shown in red color (the best results within each sub-group are shown in **bold**). The methods marked with a "⋆" are used for the ablation study of Section 4.3.

| | | MAJ | MAJ⋆-E | MAJ⋆ | MMCE | MMCE⋆-E | MMCE⋆ | Snorkel | MeTaL | LIA-E | LIA | LIA-ML |
|---|---|---|---|---|---|---|---|---|---|---|---|---|
| | | \multicolumn{11}{c}{**Accuracy (%)**} | | | | | | | | | |
| BB | 2 | 63.9±0.5 | 64.4±0.5 | **65.1**±0.7 | **66.9**±0.7 | 63.2±0.4 | 63.2±0.4 | **63.8**±0.7 | 63.0±0.7 | 65.1±0.6 | **71.5**±0.4 | — |
| | 5 | **71.1**±0.4 | 70.9±0.5 | 70.9±0.5 | 73.9±0.5 | **74.2**±0.4 | 73.2±0.5 | **72.4**±0.6 | 71.5±0.6 | 73.0±0.4 | **76.2**±0.5 | — |
| | 10 | 74.4±0.4 | **75.9**±0.5 | 75.4±0.5 | 76.5±0.3 | 76.9±0.4 | **77.0**±0.3 | 76.0±0.4 | **83.0**±0.3 | 78.1±0.4 | **83.1**±0.2 | — |
| | 20 | **76.4**±0.3 | 76.2±0.3 | 76.1±0.2 | **78.5**±0.4 | 77.7±0.4 | 77.7±0.4 | 76.0±0.3 | **87.0**±0.1 | 77.2±0.3 | **90.0**±0.3 | — |
| | 39 | 75.9±0.0 | **78.4**±0.2 | 78.4±0.0 | **79.6**±0.1 | 78.8±0.3 | 78.4±0.0 | 76.0±0.0 | **89.0**±0.0 | 78.9±0.0 | **93.0**±0.6 | — |
| WS | 2 | 82.8±0.8 | 87.2±0.8 | **87.7**±0.7 | **80.1**±0.5 | 79.3±0.4 | 80.0±0.4 | **76.2**±0.7 | 76.0±1.0 | 87.7±0.6 | **88.7**±0.7 | — |
| | 5 | 87.1±0.6 | **91.4**±0.4 | 91.3±0.4 | **87.0**±0.3 | 84.4±0.3 | 84.0±0.3 | 76.3±0.6 | **85.1**±0.7 | 87.7±0.5 | **92.7**±0.4 | — |
| | 10 | 88.6±0.2 | **93.3**±0.0 | 93.3±0.0 | **90.2**±0.3 | 80.0±0.0 | 80.0±0.0 | 76.3±0.1 | **93.1**±0.0 | 87.7±0.0 | **96.3**±0.1 | — |
| RTE | 2 | 72.8±0.2 | 72.8±0.4 | **74.5**±0.3 | 75.3±0.3 | **77.3**±0.2 | 73.2±0.3 | **65.2**±0.3 | 61.0±0.3 | 76.7±0.3 | **78.0**±0.3 | — |
| | 5 | **84.8**±0.1 | 84.0±0.2 | **84.8**±0.1 | 88.5±0.3 | **88.9**±0.2 | 85.3±0.2 | **79.1**±0.1 | 72.4±0.3 | 84.3±0.1 | **89.1**±0.1 | — |
| | 10 | 90.0±0.1 | **90.4**±0.1 | 89.9±0.1 | **92.7**±0.1 | 92.7±0.2 | 87.1±0.1 | **90.0**±0.0 | 78.0±0.0 | 91.6±0.0 | **93.1**±0.1 | — |
| MC | 2 | **26.8**±0.1 | 24.2±0.2 | 26.5±0.1 | 29.1±0.2 | **29.3**±0.2 | 22.0±0.2 | **27.1**±0.1 | 25.3±0.3 | 25.0±0.2 | 29.5±0.1 | **30.1**±0.1 |
| | 5 | 24.1±0.1 | 23.6±0.1 | **24.2**±0.1 | 24.5±0.1 | **24.6**±0.8 | 21.3±0.1 | **24.0**±0.1 | 21.0±0.1 | 24.0±0.1 | 30.9±0.3 | **36.4**±0.2 |
| | 10 | **24.1**±0.1 | 23.6±0.1 | 24.1±0.1 | 24.4±0.1 | **24.6**±0.1 | 20.1±0.1 | **24.0**±0.0 | 20.0±0.0 | 23.6±0.1 | 30.5±0.2 | **34.1**±0.3 |
| MT | 2 | 33.8±0.4 | **35.7**±0.4 | 34.2±0.2 | 35.3±0.3 | **38.6**±0.3 | 34.2±0.4 | **33.3**±0.1 | 22.1±0.3 | 34.0±0.3 | 38.6±0.4 | **40.8**±0.3 |
| | 5 | **34.2**±0.3 | 34.1±0.3 | 33.6±0.1 | 36.8±0.2 | **37.0**±0.2 | 35.0±0.4 | **34.0**±0.3 | 21.0±0.3 | 33.0±0.2 | 38.3±0.2 | **46.1**±0.5 |
| | 10 | 34.2±0.2 | 34.3±0.3 | **35.2**±0.2 | **38.5**±0.1 | 38.3±2.4 | 36.3±0.3 | **35.0**±0.1 | 03.1±0.1 | 33.8±0.2 | 42.1±0.2 | **45.4**±0.4 |

*(Row label on the left: "Dataset & Redundancy")*

Aside from these baselines, we also perform experiments using the following custom methods that we designed for the purposes of performing an ablation study (the study is presented in Section 4.3):

- **LIA-E**: In order to evaluate the usefulness of instance features, we learn embeddings of size 16 for the instances instead of using their features.
- **MAJ⋆-E**: A two-step method that resembles how machine learning models are currently being trained when using crowdsourced data. First, we estimate ground truth using **MAJ**. Next, we train the $h_\theta$ model used in **LIA-E** directly on the aggregated labels.
- **MAJ⋆**: Same as **MAJ⋆-E**, except that we use the model $h_\theta$ of **LIA** (i.e., making use of instance features instead of learning instance embeddings).
- **MMCE⋆-E**: Same as **MAJ⋆-E**, but with **MMCE** used for label aggregation.
- **MMCE⋆**: Same as **MAJ⋆**, but with **MMCE** used for label aggregation.

During each M-step we use the AMSGrad optimizer (Reddi et al., 2018) to maximize the log-likelihood function with the learning rate set to 0.001, and we perform 1,000 optimization iterations using a batch size of 1,024. Overall, we perform 10 EM iterations (all models did converge within that limit) with warm starting (i.e., the model parameters are always initialized to the values obtained during the previous M-step). When using **LIA** with image instances we use as image features the activations of the last layer of a pre-trained ResNet-101 Convolutional Neural Network (CNN). Similarly, for all text instances we use as text features the representations provided by a pre-trained BERT model (Devlin et al., 2018). More details on our setup and the model hyperparameters can be found in our code repository at `http://anonymous`.

We evaluate all methods by computing the *accuracy* of the predicted instance labels. This is a common metric for evaluating crowdsourcing methods and it also implicitly measures the quality of the confusion matrices predicted by our model. This is because these confusion matrices heavily influence the supervision provided to the ground truth model, $h_\theta$, while training. Furthermore, instead of just computing accuracy for the full datasets, we also measure how performance varies as a function of *redundancy*—the maximum number of annotations provided per instance. In order to limit the redundancy for existing datasets we randomly sample subsets of the provided annotations. Performing well in low redundancy settings is very important because it can result in significantly reduced crowdsourcing costs.

## 4.2 RESULTS

Our results are presented in Table 2. **LIA** methods consistently outperform alternative approaches. In certain cases (e.g., in Blue Birds) we are able to boost accuracy over the best alternative method by 14%, thus establishing a new state-of-the-art for this dataset. In the multi-task setting, where we train the **LIA-ML** model to jointly infer ground truth for both Medical Causes (MC) and Medical Treats (MT)

while sharing the representations of instance difficulties and annotator competencies. We observe that multi-task training boosts performance by more 8% absolute (or over 20% relative) over the single task counterpart, outperforming the baselines by over 25% relative. Finally, our approach can obtain the performance of the best alternative method using up to 4 times less redundancy, which can have significant implications for the cost of crowdsourcing, especially when annotation requires domain expertise (e.g., in healthcare). We note that **Snorkel** and **MeTaL** tend to perform well overall, but sometimes fail entirely (often performing on par with or worse than majority voting).[7] **MeTaL** also suffers from calibration issues, as it often achieves very low accuracy while having reasonable mean average precision. Data programming systems could thus benefit significantly by integrating our method in their pipeline, tying together the label aggregation and model training phases.

## 4.3 ABLATION STUDY

Our main contributions are: (i) end-to-end learning by fusing the label aggregation and model training phases, and (ii) allowing for instance and annotator features to inform label aggregation.[8] In this section, we show how each one of these contributions is important on its own by performing experiments where we introduce each one on their own, while keeping everything else constant.

**Table 3:** Results for **LIA** without marginal likelihood fine-tuning.

| Dataset & Redundancy | | Accuracy (%) |
|---|---|---|
| BB | 2 | 71.9±0.9 |
| | 5 | 74.9±0.4 |
| | 10 | 76.9±0.5 |
| | 20 | 77.5±0.4 |
| | 39 | 79.1±0.2 |
| WS | 2 | 88.7±0.8 |
| | 5 | 91.0±0.6 |
| | 10 | 93.3±0.0 |
| RTE | 2 | 63.8±0.1 |
| | 5 | 67.5±0.1 |
| | 10 | 69.6±0.1 |
| MC | 2 | 24.0±0.2 |
| | 5 | 23.9±0.2 |
| | 10 | 23.0±0.2 |
| MT | 2 | 38.1±0.3 |
| | 5 | 38.5±0.2 |
| | 10 | 39.2±0.1 |

**End-to-End Learning.** The best way to test the effectiveness of end-to-end learning is to compare end-to-end approaches with two-stage approaches where: (i) we first aggregate labels, and (ii) we then train machine learning models using the aggregated labels. To this end, we introduced the baseline methods marked with a "⋆" in Table 2. The results indicate that the two-stage approach underperforms the base label aggregation method for both **MAJ** and **MMCE**. This is most likely due to the fact that in both these cases the model being learned cannot inform the label aggregation stage. In contrast, **LIA** is able to outperform all two-stage approaches because it allows for exactly that. Note that **MMCE** models instance difficulty and annotator competence similar to **LIA**, with the exception that it does not use instance features and it does not allow for end-to-end learning of the ground truth predictor. Also note that **LIA-E** does not use instance features, but is still able to outperform **MAJ⋆-E** and **MMCE⋆-E** in many cases, indicating that end-to-end learning is, in fact, effective and accounts for at least part of the performance gains achieved by **LIA**.

**Instance Features.** The methods which use instance features are **LIA**, **MAJ⋆-M**, and **MMCE⋆-M**. To test for the usefulness of these features, we provide variants of these methods (labeled **LIA-E**, **MAJ⋆-E**, and **MMCE⋆-E**, respectively, in Table 2) that use indicator features instead and learn instance embeddings. We observe that for **MAJ** and **MMCE** the results are inconclusive (the feature-based methods outperform the alternative in around half of the experiments). However, in the context of end-to-end learning, we observe that **LIA** consistently outperforms **LIA-E** by a significant margin. This indicates that instance features are indeed useful, especially so in the context of end-to-end learning where they can inform the label aggregation phase.

It is important to also mention that results pertinent to this ablation study for the Word Similarity (WS) dataset were a bit unstable with many models effectively failing to learn anything meaningful (specifically **MMCE⋆-E**, **MMCE⋆-M**, and **LIA-E**). Our best explanation for this is that the Word Similarity (WS) dataset is very small with only 30 instances and is thus highly prone to overfitting.

Finally, in order to evaluate the effect of the marginal likelihood fine-tuning approach presented in Section 3.1, we also run experiments using **LIA** without this fine-tuning phase. The results are shown in Table 3 and it is clear that marginal likelihood fine-tuning results in better performance. Also, in order to sanity check that **LIA** is able to predict the qualities of the predictors accurately, we also performed a synthetic experiment. More specifically, we added an "always correct" and an "always wrong" oracle to all datasets used in our experiments. It turns out the predicted qualities for the two oracles are the highest and lowest among all predictors, respectively. This indicates that our model is indeed capable of uncovering such highly competent and incompetent predictors.

---

[7]This behavior has also been observed by others (e.g., `https://github.com/HazyResearch/snorkel/issues/1073`).

[8]As mentioned in the beginning of this section, we were unable to obtain any crowdsourcing datasets with associated annotator meta-data and thus in our experiments we only use instance features.

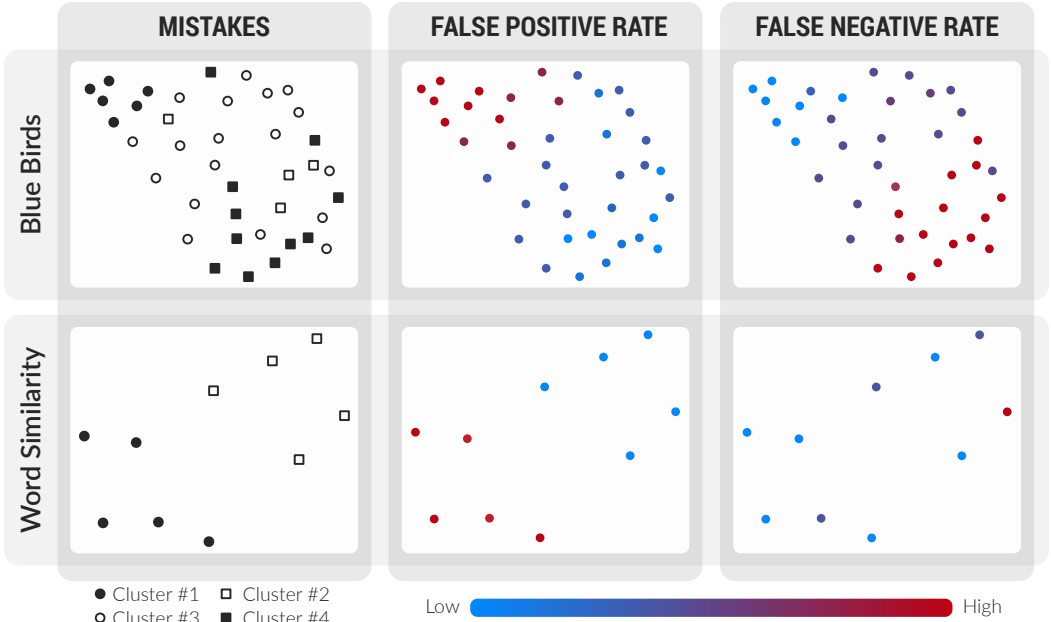

**Figure 2:** Visualization of the learned predictor embeddings. Each dot in the figures represents a predictor projected on 2D plane using UMAP. One the left, the predictors were first clustered based on which instances they make mistakes on and then colored and shaped based on which cluster they belong to (in the embedding space, predictors that make similar mistakes tend to cluster together). In the middle, the predictors are colored based on their false positive rate. On the right, they are colored based on their false negative rate.

### 4.4 PREDICTOR EMBEDDINGS VISUALIZATION

To evaluate whether the learned predictor embeddings are meaningful in some way, we perform dimensionality reduction using UMAP (McInnes et al., 2018), plot them in Figure 2, and color predictors in three different ways, which lets us understand the information captured by the manifold:

1. Mistakes Cluster: To cluster predictors, we represent each with a one-hot vector that indicates the instances it made mistakes on, and then run agglomerative clustering with $L_1$ distance metric. On the plot, each cluster is associated with a unique shape.
2. False Positive Rate: Each predictor is colored based on its false positive rate.
3. False Negative Rate: Each predictor is colored based on its false negative rate.

We have provided figures for Blue Birds and Word Similarity datasets which are the only ones for which all predictors annotated all instances (it is unclear how to properly compute the mistakes clustering distance metric when some or most annotations are missing). From these plots, it is clear that the learned predictor embeddings encode both the expertise of the corresponding predictor as well as the likelihood of making a false positive or false negative mistake.

## 5 CONCLUSION

In this paper, we have introduced a learning framework for: (i) training deep models directly on data with imperfect annotations, and (ii) modeling the processes that produced the labels. Our approach improves upon the classical and widely used two-stage setup (first aggregate and denoise the labels, then train the model) by merging the two stages. As a result, we are able to train models end-to-end using multiple noisy labels, while estimating the difficulties of the examples and learning accurate representations for the annotators that produced the labels. Experimental results on multiple small and large scale publicly available crowdsourcing datasets indicate that our method results in significant gains in accuracy (up to 25% relative gain over the current state-of-the-art approaches for aggregating noisy labels). Moreover, it turns out that training the model to predict multiple related labels simultaneously improves the learned representations and results in further gains in predictive performance of the model. Finally, we performed an ablation study to evaluate the effect of both end-to-end learning and instance features and showed that both contribute to the performance gains achieved by the proposed method.

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
