# OpenReview forum: "Learning from Imperfect Annotations: An End-to-End Approach"
_ICLR.cc/2020/Conference — Reject_

### Official Review · AnonReviewer1 · 2019-10-22
**Official Blind Review #1**

**Rating:** 6

**Review:**

This paper proposes a method for dealing with noisy human annotated in training data. The idea is to unify the annotation aggregation and model training whilst modelling the sample annotation difficulty and annotator competency level. The experiments on five datasets show improvements over a number of baselines.

This is a solid piece work that deals with a very practical problem – training ML models with crowdsourced data with imperfect annotations. The proposed method is not completely new: many ideas have been adopted from previous works. However putting everything together seems to work as demonstrated by the experiments.

I have a number of concerns:

1, One of the main claimed novelties of the paper is an “end-to-end” approach that unifies ground truth label prediction and label aggregation. However there is no ablation study to show that this is indeed better than a two-stepped variant with everything else remaining the same.

2, The authors made a claim on the ability to estimate the quality of the annotator. But the statement is the Introduction is misleading – it suggests that the attributes of the annotator such as age and gender will be used as input to the estimator, but later it is clear that no benchmarks contain that information hence it is not implemented. Also it is not clear how this vector embedding (Sec. 3.2) is implemented. Any evidence that this embedding is indeed a clustering index?

3, In the implementation, the strong BERT model was used for language but VGG was used for image representation. Stronger CNN feature extractors such as ResNet101 should be used.

4, In general, the lack of any ablation study is a problem for analysing why the model works.


**Experience Assessment:**

I have published one or two papers in this area.

**Review Assessment: Checking Correctness Of Derivations And Theory:**

I assessed the sensibility of the derivations and theory.

**Review Assessment: Checking Correctness Of Experiments:**

I assessed the sensibility of the experiments.

**Review Assessment: Thoroughness In Paper Reading:**

I made a quick assessment of this paper.

---

> ### Author Response · Authors · 2019-11-10
> **Response to Official Blind Review #1**
>
> Thank you for the feedback and helpful comments! We address your concerns below.
>
> 1. Lack of end-to-end vs. two-step ablation:
>
> This is a valid concern. To this end, we have added an entirely new section (4.3) where we perform an ablation study to try and understand which parts of the proposed model contribute to its performance gains. Overall we observe that both end-to-end learning and the use of instance features play an important role, with the former being perhaps the most valuable contribution. As part of this study we also compare with two-stage variants.
>
> 2. Annotator representations:
>
> As you point out, in the datasets considered in our study, no worker features were available besides their IDs. In fact, we were unable to find datasets where the worker features were available (partially because no prior models were able to use that information). Hence we opted to model workers with embeddings. If worker features were available, our framework could directly accommodate for that. We have added a brief note on that in the current paper revision. Regarding the learned embeddings behaving as a clustering index, we have also added a couple of visualizations that showcase this point (Section 4.4 in the revised version).
>
> 3. ResNet101 feature extractors:
>
> This is a valid point and we thank you for your suggestion. We actually re-ran all image-based experiments using a pre-trained ResNet101 feature extractor and have already updated the submitted manuscript. This change resulted in even more significant performance gains for our method (e.g., our results for BlueBirds shown in Table 2 are now significantly stronger).
>
> 4. Ablation study:
>
> We addressed this concern by adding an ablation study section (4.3).

---

### Official Review · AnonReviewer3 · 2019-10-24
**Official Blind Review #3**

**Rating:** 6

**Review:**

Update after author response:
I would like to thank the authors for their thoughtful response, and for addressing some of the concerns raised by the reviewers. One of my main complaints arose from a misunderstanding that none of the baselines model worker competencies and task difficulty. The authors clarified that MMCE does model those, making it a competitive baseline. Given that and the other additions to the draft, I am changing my assessment from 3 to 6.
---------------------------

In this paper, the authors extend the classical probabilistic model of Dawid-Skene (DS) for predicting the final label of crowdsourced tasks. The extensions include explicit modeling of image difficulty and worker competence as a function of image and worker features respectively, as well as a more expressive formulation of learned functions in terms of neural networks (NNs). The authors show that the proposed approach outperforms several baselines on 5 different datasets.

The problem studied in the paper is relevant since more and more data is needed to train ever more complex models, and often crowdsourcing is the way to generate such data. The paper is clearly written and conveys its central idea concisely. I liked the paper but I believe the contribution to be incremental in its current state. Here are the main issues in my opinion:

1. The presentation suggests that somehow the proposed approach is novel in its end-to-end framework. However, the central idea is very similar to Dawid-Skene (1979), and its subsequent augmentations like the ones in Carpenter (2008) that model both image difficulty and worker competence. These previously proposed models are also end-to-end approaches so that they can infer worker competence and image difficulties while also outputting a final label. Therefore, I believe the novel contributions are then slightly different formulation of these variables, and using NNs to learn the function parameters. I see that the proposed model can be somewhat more general since it uses semantic image features (as opposed to indicator features) but then it learns worker embeddings starting from random initializations which will not generalize to new workers, necessitating a re-run of the whole EM loop.

2. If Q is a stochastic matrix, its rows should sum to 1. I don’t understand the summation on Q in eq. 4. I am also a little confused by eq. 5 since the joint likelihood should just be the prior times the conditional likelihood of the observed annotations. At least trying to do it in my head doesn’t lead to what’s presented in eq. 5. It may just be an issue of notations which can be simplified by using conditional distributions instead of expectations of indicator random variables (same quantity).

3. As far as I understand, none of the baselines considered explicitly model the task difficulty and worker competence. Therefore, the proposed model enjoys this extra level of expressiveness making its superior performance relatively unsurprising. I skimmed Zhou et al. (2015) and simple DS was already competitive on some of the datasets so I would think inclusion of its successors that model image difficulty and worker competence could perform quite well.

3. Minor issue but eq. 6 is not majority vote as stated just above. It’s the maximum likelihood estimate P(y_i = k | D) assuming each worker is an unbiased estimator of the true label.

4. Minor formatting issues: “learn a models” (sec 3.2), “lantent variables” (sec 4) “and by merging” (sec 5).

In summary, even though I like that the paper is clearly written and tackles an important problem, I am not convinced that the contribution is substantial or the experiments very insightful. I would recommend addressing these issues and resubmitting the paper.

Carpenter, B. (2008). Multilevel bayesian models of categorical data annotation. Unpublished manuscript, 17(122), 45-50.

**Experience Assessment:**

I have read many papers in this area.

**Review Assessment: Checking Correctness Of Derivations And Theory:**

I assessed the sensibility of the derivations and theory.

**Review Assessment: Checking Correctness Of Experiments:**

I carefully checked the experiments.

**Review Assessment: Thoroughness In Paper Reading:**

I read the paper thoroughly.

---

> ### Author Response · Authors · 2019-11-10
> **Response to Official Blind Review #3**
>
> Thank you for the feedback! We address your comments and questions below.
>
> 1.a. End-to-End Learning:
>
> Thanks for raising this point. We agree that Dawid-Skene (1979) and the follow-up work are all end-to-end in the sense that they simultaneously infer the ground truth along with instance difficulties and worker competencies. However, once the ground truth is obtained the dataset is typically fixed and used for downstream training. Previous work (implicitly) separates ground truth inference (i.e., label aggregation) and model training. More specifically, in prior work one would train a model that predicts ground truth in 2 stages: (i) infer the ground truth labels from the provided annotations, and (ii) train a machine learning model on the inferred labels. Our approach merges these two stages and allows us to train machine learning models directly on the imperfect annotations. This is what we mean by end-to-end, which is different from the meaning you suggested and it is particularly important *because the assumptions made by the machine learning model (i.e., inductive biases) can now be taken into account when aggregating the imperfect annotations*. We have revised the paper to clarify/emphasize this point (last paragraph of Section 2) as well as ablate end-to-end vs. two-stage training in the newly added Section 4.3.
>
> 1.b. Worker Embeddings:
>
> Unfortunately in the datasets considered in our study, no such features were available besides annotator IDs. In fact, we were unable to find datasets where the annotator features were available (partially because no prior models are able to use this information). Hence we opted to model workers with embeddings. If worker features were available, our framework could directly accommodate for that. We have added a brief note on that in the current paper revision.
>
> 2. Clarification of Eqs. 4-5:
>
> Q is indeed a stochastic matrix and its rows sum to 1. However, in eq. 4 we are dividing by the sum of each column, thus making the columns sum to 1. This is necessary in order to compute the probability that the true label is equal to k given the annotator labels, as opposed to the reverse, which is what each row of Q represents. Regarding the derivation of eq 5., let us try to clarify. The full likelihood of the data given the $y_i$’s (which are latent variables) is defined as $\mathcal{L} = \prod_{i=1}^N p(y_i) \prod_{j\in\mathcal{M}_i} p(\hat{y}_{ij} \mid y_i)$. In the E-step we compute the expectation of the latent variables, which we denote by $\tilde{y}_i$, for each $i$. Then, during the M-step we maximize this likelihood function with $y_i = \tilde{y}_i$, which results in:
>
> $$\begin{align}
>   \mathcal{L} &= \prod_{i=1}^N p(\tilde{y}_i) \prod_{j\in\mathcal{M}_i} p(\hat{y}_{ij} \mid \tilde{y}_i), \\
>   \log\mathcal{L} &= \sum_{i=1}^N \log p(y_i = \tilde{y}_i) + \sum_{i=1}^N \sum_{j\in\mathcal{M}_i} \log p(\hat{y}_{ij} \mid y_i = \tilde{y}_i), \\
>   \log\mathcal{L} &= \sum_{i=1}^N \sum_{k=1}^C \tilde{y}^k_i \log [h_{\theta}(x_i)]_k + \sum_{i=1}^N \sum_{k=1}^C \tilde{y}^k_i \sum_{j\in\mathcal{M}_i} \log \frac{[\mathbf{Q}_{ij}]_{k\hat{y}_{ij}}}{\sum_{l=1}^C [\mathbf{Q}_{ij}]_{l\hat{y}_{ij}}}.
> \end{align}$$
>
> We have clarified the derivation in the paper.
>
> 3.a. “None of the baselines considered explicitly model the task difficulty and worker competence.”
>
> This is not true. The MMCE baseline (Zhou et al., 2015) does explicitly model task difficulty and worker competence. However, similar to Dawid-Skene (1975) it is not able to take into account features about the data instances or the workers and uses indicator features instead. In fact, even though we use an entirely different formulation from Zhou et al., our approach generalizes their method (replacing all features with indicator features and removing the use of neural networks results in MMCE). Given that MMCE consistently outperforms Dawid-Skene we compare against that approach to show how the use of data features and thee end-to-end training of neural networks is what actually results in a significant performance boost, rather than just the extra level of expressiveness due to the explicit modeling of task difficulty and worker competence. We have now also updated the paper to include an entirely new section (4.3) where we perform an ablation study to try and understand which parts of the proposed model contribute to its performance gains.
>
> 3.b. Majority Vote Clarification:
>
> Eq. 6 is indeed the maximum likelihood estimate $p(y_i = k \mid \mathcal{D})$ assuming each worker is an *independent* unbiased estimator of the true label. This matches our definition of a majority vote, but please let us know if you have a different definition in mind and we will be happy to clarify.
>
> 4. Formatting Issues:
>
> Thank you for being thorough and pointing out these issues. We have now corrected them.
>
> We have revised the manuscript to address your concerns and hope that we have clarified the significance of our contribution.

---

### Official Review · AnonReviewer4 · 2019-11-03
**Official Blind Review #4**

**Rating:** 6

**Review:**

The submission addresses a problem with collecting ground truth: human annotations are noisy, a common approach is to collect many annotations and apply majority voting to elicit a single label. With this approach, the annotators' expertise and the difficulty of single data instances is ignored. What the authors propose is a framework which allows one to combine a direct graphical model of how human annotations are produced with model training. The graphical model introduces latent variables for the difficulty of an instance and the competence of an annotator as well as the (unobserved) true label. This way one can potentially benefit from the meta-information about the annotators (e.g., their demographics) and improve upon the majority-voting baseline of aggregating available annotations. Maybe most importantly, the proposed framework allows one to get the true label with fewer annotations (significantly reduces redundancy).

The proposed model is intuitive, the learning of the hidden variables is done with EM, the presentation is easy to follow. The experimental part is done on five annotation tasks (image classification, NLP, bio-NLP) and compares the proposed model (LIA) with six prior approaches (e.g., majority vote, MMCE, Snorkel). The evaluation metric is accuracy (i.e., guessing the true label as obtained from experts). Overall, the new model achieves higher or comparable accuracy to that of MMCE which in turn outperforms all other methods. W.r.t. redundancy reduction, on some tasks LIA achieves much better accuracy with fewer annotations. For example, on the word similarity task the accuracy with only two annotations is higher than that of the majority baseline with ten.

The submission is well-written and I enjoyed reading it. I am not an expert in this area, but as far as I am concerned the contributions are sufficient to accept it. I have not found any technical or methodological flaws. However, I have the following questions and concerns:

1. The analysis part is very short and while the accuracy numbers are impressive, I wonder if a different experiment is needed to fully demonstrate the claimed benefits of the model. For example, I am not sure which part shows empirically that annotators' features are indeed used and useful.

2. Related to the above point, maybe one could create a synthetic dataset where the true labels and true noise are added as if coming from two additional annotators and show that the model can identify them as highly competent / incompetent?

3. Section 3.1 mentions a fine-tuning procedure in the end but the experimental part does not specify how much of a gain it delivered. How does the model perform without this fine-tuning?

4. I have not followed this topic much but it seems to me that there should be more related work on modelling annotators' competence and item difficulty for crowd-sourced annotations. Isn't, for example, work by Bachrach et al. 2012 [1] relevant?

5. How stable are results along the redundancy dimension? For example, the word similarity task has only 30 word pairs with ten ratings per item. How much, if at all, is accuracy with redundancy@2 affected by using different samples of two?

Minor typos:
 - "can be can be" in Sec. 1 on page 2.
 - "a models" in Sec. 3.2. on page 5.
 - "a sources" in Sec. 3.3 on page 6.
 - "LIA-E" (should be "LIA"?) on page 7.

[1] "How To Grade a Test Without Knowing the Answers — A Bayesian
Graphical Model for Adaptive Crowdsourcing and Aptitude Testing" ICML'12.

**Experience Assessment:**

I do not know much about this area.

**Review Assessment: Checking Correctness Of Derivations And Theory:**

I carefully checked the derivations and theory.

**Review Assessment: Checking Correctness Of Experiments:**

I assessed the sensibility of the experiments.

**Review Assessment: Thoroughness In Paper Reading:**

I read the paper at least twice and used my best judgement in assessing the paper.

---

> ### Author Response · Authors · 2019-11-10
> **Response to Official Blind Review #4**
>
> Thank you for the encouraging feedback! We address your comments and questions below.
>
> 1. “The analysis part is very short.”
>
> This is a valid concern that was shared among all reviewers. To address it, we have added an entirely new section (4.3) where we perform an ablation study to try and understand which parts of the proposed model contribute to the performance gains. Overall, we observe that both end-to-end learning and the use of instance features play an important role, with the former being perhaps the most valuable contribution. Regarding annotator features, unfortunately in the datasets considered in our study, no such features were available besides annotator IDs. In fact, we were unable to find datasets where the annotator features were available (partially because no prior models were able to use this information). Hence we opted to model workers with embeddings. If worker features were available, our framework could directly accommodate for that. We have added a brief note on that in the current paper revision.
>
> 2. “Related to the above point, maybe one could create a synthetic dataset where the true labels and true noise are added as if coming from two additional annotators and show that the model can identify them as highly competent / incompetent?”
>
> This is an interesting idea, thank you for the suggestion! During the rebuttal period we tried adding an “always correct” oracle and an “always wrong” oracle to the existing datasets that we used in our experiments. It turns out the predicted accuracies for the two oracles are the highest and the lowest among all predictors, respectively. This indicates that our model is indeed capable of uncovering such highly competent and incompetent predictors. We have added a brief mention of this experiment at the end of Section 4.3. In addition to that, we also visualized the learned embeddings for each predictor in the BlueBirds dataset, which shows that the learned representations indeed capture some structure that correlates with predictor competencies (Section 4.4 in the revised version).
>
> 3. “Section 3.1 mentions a fine-tuning procedure in the end but the experimental part does not specify how much of a gain it delivered. How does the model perform without this fine-tuning?”:
>
> We have added Table 3 (as part of the new ablation study section) which contains the results obtained for LIA when not using the marginal likelihood fine-tuning procedure. From these results it is clear that using this procedure results in a significant performance gain. We also now mention this in the end of section 4.3.
>
> 4. “Related work on modeling annotators’ competence.”
>
> Thank you for pointing out the work by Bachrach et al. (2012). We have now included this and a couple other related publications in the updated manuscript, along with a discussion on how our work differs. Note also that MMCE (one of our baselines, Zhou et al., 2015) also attempts to model item difficulty and annotator competence, but it lacks the ability to perform end-to-end learning and use instance/annotator features. We believe that to be the main reason our method outperforms MMCE.
>
> 5. “How stable are the results along the redundancy dimension?”
>
> This is a good question. To produce each number in Table 2, we ran experiments 50 times using different random seeds, which resulted in different subsets of redundant annotations being selected for each instance at each run. The standard error (shown in gray color next to each number) captures the level of stability. Accuracy is of course affected by selecting different subsets (typically affected more in lower redundancy regimes, as shown in Table 2), but the effect is marginal compared to the gains achieved by LIA, which also implies statistical significance of our reported results.
>
> Thank you for being thorough and pointing out some typos. We have corrected them in the updated manuscript.

---

### Author Response · Authors · 2019-11-10
**Summary of Response to Reviews**

We thank all reviewers for the helpful feedback and comments!

We respond to each one individually, but we would also like to emphasize that in order to address some of the concerns that were shared across multiple reviewers, we added Sections 4.3 and 4.4. Section 4.3 contains an ablation study of how each part of the proposed model contributes to the observed performance gains. Overall, we observe that both end-to-end learning and the use of instance features play an important role, with the former being perhaps the most valuable contribution. As part of this study, we also compare against two-stage variants where the annotations are first aggregated to form ground truth label estimates and then these estimates are used to train machine learning models. This, along with a partial rewrite of parts of our experiments section should provide a more extensive analysis of why LIA works well and which of our contributions are most important in achieving performance gains. Section 4.4 includes a visualization of the learned predictor embeddings that showcases how they contains useful information. Finally, we have also clarified what we mean by “end-to-end learning” and also how annotators are being modeled in our experiments.

---

### Decision · Program_Chairs · 2019-12-19

**Decision:**

Reject

**Comment:**

The paper introduces a novel way of jointly modeling annotator competencies and learning from imperfect annotations. Reviewers were moderately positive. One reviewer mentioned Carpenter (2002) and subsequent work. One prominent example of this line of work, which the authors do not cite, is: https://www.isi.edu/publications/licensed-sw/mace/ - from 2013. I encourage the authors to cite this paper. In the discussion, the authors point out this type of work is not *end-to-end* in their sense. However, there's, to the best of my knowledge, a relatively big body of literature on end-to-end approaches that the authors completely ignore, e.g., [0-3]. In the absence of a discussion of this work, it is hard to accept the paper.

[0] https://link.springer.com/article/10.1007/s10994-013-5411-2
[1] https://ieeexplore.ieee.org/stamp/stamp.jsp?arnumber=7405343
[2] http://www.cs.utexas.edu/~atn/nguyen-acl17.pdf
[3] https://arxiv.org/pdf/1803.04223.pdf

---

> ### Author Response · Authors · 2019-12-21
> **Thank you for the meta-review and for bringing up some of the citations we missed**
>
> We will certainly incorporate a discussion of the suggested work (of which we were previously unaware of). We did not intentionally “completely ignore” these papers, as the AC seems to imply. Moreover, it is frustrating that the AC is bringing up the lack of some references in their meta-review along with the decision -- that could have been brought to our attention during the discussion phase and would have been easily addressed similar to how we addressed the other reviewers’ concerns. We put a lot of effort into adding 2 pages of actual content in order to fully address all the reviewers’ concerns during the rebuttal, and so we believe that getting a rejection over a couple more missing citations without having a chance to address them is disappointing and unfair.
>
> After carefully examining the suggested references, we do agree that those papers propose different end-to-end architectures for learning directly from crowdsourced data, but our approach is significantly different. The suggested references are indeed related work, but this fact neither affects the novelty nor the key methodological and substantial experimental contributions of our work (with 25% improvements over the state of the art for some of the datasets).